# Investigation of Protein Corona Formed around Biologically Produced Gold Nanoparticles

**DOI:** 10.3390/ma15134615

**Published:** 2022-06-30

**Authors:** Parastoo Pourali, Eva Neuhöferová, Volha Dzmitruk, Veronika Benson

**Affiliations:** 1Institute of Microbiology, Czech Academy of Sciences, 142 20 Prague, Czech Republic; parastoo.pourali@biomed.cas.cz (P.P.); neuhoferova.eva@gmail.com (E.N.); 2Center of Molecular Structure, Institute of Biotechnology, Czech Academy of Sciences, 252 50 Prague, Czech Republic; volha.dzmitruk@ibt.cas.cz

**Keywords:** biologically produced gold nanoparticles, hard protein corona, capping agent, *Fusarium oxysporum*

## Abstract

Although there are several research articles on the detection and characterization of protein corona on the surface of various nanoparticles, there are no detailed studies on the formation, detection, and characterization of protein corona on the surface of biologically produced gold nanoparticles (AuNPs). AuNPs were prepared from *Fusarium oxysporum* at two different temperatures and characterized by spectrophotometry, Fourier transform infrared spectroscopy (FTIR), transmission electron microscopy (TEM), and energy-dispersive X-ray spectroscopy (EDS). The zeta potential of AuNPs was determined using a Zetasizer. AuNPs were incubated with 3 different concentrations of mouse plasma, and the hard protein corona was detected first by sodium dodecyl sulfate-polyacrylamide gel electrophoresis (SDS-PAGE) and then by electrospray liquid chromatography–mass spectrometry (LC-MS). The profiles were compared to AuNPs alone that served as control. The results showed that round and oval AuNPs with sizes below 50 nm were produced at both temperatures. The AuNPs were stable after the formation of the protein corona and had sizes larger than 86 nm, and their zeta potential remained negative. We found that capping agents in the control samples contained small peptides/amino acids but almost no protein(s). After hard protein corona formation, we identified plasma proteins present on the surface of AuNPs. The identified plasma proteins may contribute to the AuNPs being shielded from phagocytizing immune cells, which makes the AuNPs a promising candidate for in vivo drug delivery. The protein corona on the surface of biologically produced AuNPs differed depending on the capping agents of the individual AuNP samples and the plasma concentration.

## 1. Introduction

Nanoparticles have special properties due to their larger surface area compared to their bulk material [1]. Due to their small size, they can easily interact with their environment and enter the body through different routes, such as inhalation, absorption through the skin, or ingestion [2]. Sometimes, some of them are injected into the body for biomedical purposes [2]. When they enter the human body unintentionally or intentionally, they are surrounded by bodily fluids immediately after internalization [2]. The surface of the nanoparticles is then covered by various types of macromolecules called “soft corona”, which is mostly composed of proteins. Over time, the soft corona is displaced by the “hard corona”, which has a higher binding affinity [3,4,5]. It has been widely reported that the type of macromolecules surrounding the nanoparticles, as well as the type and nature of the nanoparticles, such as surface charge, size, shape, solubility, and other physicochemical properties, are important for the composition of the corona [6,7].

Among the various nanoparticles used in the biomedical field, gold nanoparticles (AuNPs) are attracting more attention due to their biocompatibility, low toxicity, and most importantly, their high drug delivery capacity [8,9]. They are also used as tools for gene or molecule delivery, imaging, biosensors, and hyperthermia [10]. The main method for producing AuNPs is a chemical technique [11], but it has some limitations, such as the fate of toxic chemical reagents on the surface of the nanoparticles or their release into the environment [1]. Another production method that is currently being tested is the biological method [12,13]. In this technique, the nanoparticles are produced enzymatically or/and non-enzymatically, and some microbial strains (i.e., bacteria and fungi) [14] or some plant extracts [15] are used to reduce the toxic metallic ions into less toxic and elemental forms [11,16]. Different reports showed that fungal strains, such as *Verticillium* sp. [17], *Trichothecium sp.* [18], *Aspergillus niger* NCIM 616 [19], and *Fusarium oxysporum* [20] can produce AuNPs covered by microbial or plant proteins. Those molecules are commonly called “capping agents” [1,8,21]. Although some physical parameters such as pH are important for nanoparticle stability [8], the capping agents are responsible for separating and stabilizing the nanoparticles so that they do not agglomerate in close contact [1,8,10]. The capping agents have been shown to be proteins [21], but other biomolecules such as polysaccharides, vitamins, alkaloids, phenols, terpenoids, co-enzymes, carbohydrates, enzymes, etc. can also act as both reducing and capping agents to cover the surface of the nanoparticles, so the nature of the capping agents varies from one species to another [22]. Two proteins of 25 kDa and 19 kDa have been reported to serve as capping agents for AuNPs produced by *F. oxysporum* [23]. There are examples that we mention in the Discussion section.

In our previous studies, Fourier transform infrared spectroscopy (FTIR) has been applied to detect the protein structure, capping agents, on the surface of AuNPs [8,10,24] that were produced by *F. oxysporum*. The capping agents on the surface of biologically produced AuNPs were able to bind to various cargoes, such as some antibiotics [24,25], without external binders. Therefore, the AuNPs are likely to bind to various proteins in serum once they enter the body, and characterization of the protein corona is important regarding nanoparticles’ fate in the body. It has been reported that adsorption of complement factors or IgG to the surface of nanoparticles results in their elimination from the body by phagocytosis [26,27], whereas their prior shielding with serum albumin prevents their elimination [28].

Although there are several research articles on the detection and characterization of protein corona on the surface of various nanoparticles [2,29], there are no detailed studies on the formation, detection, and characterization of protein corona on the surface of biologically produced nanoparticles. Here, AuNPs were prepared from *F. oxysporum* at two different temperatures and characterized by different techniques such as transmission electron microscopy (TEM), energy-dispersive X-ray spectroscopy (EDS), and FTIR. The zeta potential of AuNPs was determined using a Zetasizer. The nanoparticles were incubated with different concentrations of mouse plasma for 24 h. The hard protein corona was first detected by sodium dodecyl sulphate-polyacrylamide gel electrophoresis (SDS-PAGE), and then the protein composition was determined by electrospray liquid chromatography–mass spectrometry (LC-MS). In this way, the type and amount of the major proteins could be determined. For comparison, AuNPs alone were used as control.

## 2. Materials and Methods

### 2.1. Cultivation of Fungi and Preparation of AuNPs

In this study, two types of AuNPs were prepared by the biological method to compare the composition of their protein corona as well as of their capping agents. For this purpose, Sabouraud dextrose broth (SDB, Sigma-Aldrich, Prague, Czech Republic) was used to culture *F. oxysporum* (CCF 3732, Prague, Czech Republic) at 28 °C and 150 rpm for 1 week. The culture supernatant was obtained by centrifugation at 8000 rcf for 10 min [8], and AuNPs were prepared by adding HAuCl_4_·3H_2_O (Sigma-Aldrich, Prague, Czech Republic) at a final concentration of 1 mmol. The pH was adjusted to 7.5, and the mixture was divided into two flasks, one of which was incubated at 37 °C and 200 rpm for 24 h [24] (referred to here as “cold”) and the other at 80 °C for 5 min (referred to here as “hot”). Negative control vial containing sterile SDB plus 1 mmol HAuCl_4_·3H_2_O at the final concentration was subjected to the same procedures. The produced AuNPs were precipitated by centrifugation at 22,000 rcf for 20 min and washed three times with ddH_2_O [9,10,25].

### 2.2. Characterization of the AuNPs

The color-altered “cold” and “hot” sample dispersions, indicative of the preparation of AuNPs, were analyzed using the techniques listed below.

#### 2.2.1. Characterization of the AuNPs Using Visible Spectrophotometry

The maximum absorption peak of each AuNP dispersion was determined using the Nanodrop spectrophotometer (Thermo Fisher Scientific, Waltham, MA, USA). The spectra were measured in the range of 400–700 nm, and the blank sample was ddH_2_O [10,20,24].

#### 2.2.2. TEM and EDS

Samples were diluted 1/10 and 1/100 with ddH_2_O, and 2 µL of each were applied to a glow-discharge-activated (30 s, 1 kV, 10 mA) grid covered with 4 nm carbon foil and air-dried. Imaging was performed using a JEOL JEM -F 200, operated at 200 kV and equipped with a cold FEG, TVIPS XF416 camera, and JED 2300 X-ray spectrometer (JEOL, Freising, Germany) [30].

#### 2.2.3. Characterization of the AuNPs: Z-Potential and DLS Measurements

The zeta potential and size distribution of each of the “cold” and “hot” samples were determined by DLS (non-invasive backscattering technique) and ELS (electrophoretic light scattering) in aqueous solution using a Zetasizer Ultra (Malvern Panalytical, Malvern, UK). The refractive index was 0.18, the absorbance was 3.43, and the dispersant was ddH_2_O. An ultra-low-volume ZEN2112 quartz cuvette was used for size distribution analysis, and the signal was acquired in backscatter mode. A DTS1070 zeta cell with folded capillaries was used for zeta potential analysis. The set temperature was 25 °C [8].

#### 2.2.4. FTIR

Both of the AuNP samples were first freeze-dried, and then IR measurements were performed using a Vertex 70 v FTIR spectrometer (Bruker Optics GmbH, Ettlingen, Germany) with a mercury cadmium telluride (MCT) detector cooled by liquid nitrogen. The dried samples were placed in a DC3 diamond compression cell (Specac Ltd., Orpington, UK), and their IR absorption spectra were recorded against a clean DC3 cell at wavenumbers of 4000–900 cm^−1^ in a nitrogen gas atmosphere to avoid CO_2_ or water vapor peaks. The number of scans was 128–256 with a resolution of 4 cm^−1^. Spectra were recorded and processed using OPUS software version 8.5 (SP1) [8,31].

### 2.3. Preparation of the Plasma

Blood was collected from Balb/c mouse heart by cardiac puncture using an EDTA-treated syringe with a 20 G needle [32]. The collected blood was centrifuged at 800 rcf for 5 min. The supernatant was stored at −20 °C until use, and plasma was not refrozen after thawing [2]. Animals’ procedures were approved by the Ethical Committee of the Czech Academy of Sciences (under experimental plan number 82/2015) in accordance with the Czech law regarding animal protection.

### 2.4. Preparation of Hard Protein Corona

Equal amounts of AuNP samples (each contained 0.15 mg/mL Au as determined by graphite furnace atomic absorption spectroscopy (GF-AAS [33])) were centrifuged, and each pellet was dispersed in phosphate-buffered saline (PBS) at pH 7.5. Three different plasma concentrations were obtained by dilution with AuNPs to give different final concentrations of 55% *v*/*v* (corresponding to protein concentration in blood circulation), 25% *v*/*v*, and 10% *v*/*v* (corresponding to protein concentration in cell culture) plasma. For each sample, three sets were prepared and incubated at 4 °C for 24 h [34].

To obtain the hard protein corona, the samples were first centrifuged at 22,000 rcf and 4 °C for 20 min, and each pellet was washed three times with PBS. In this way, the soft protein corona was washed away, and the pellets were dispersed in PBS and stored at −20 °C before further analysis [29].

### 2.5. Detection of the Hard Protein Corona

To analyze the hard protein corona on the surface of washed AuNPs, the following methods were used:

#### 2.5.1. Detection of the Hard Protein Corona Using Visible Spectrophotometry

The maximum absorption peak of each AuNP sample changed after the formation of the protein corona. Therefore, all samples were analyzed with a Nanodrop spectrophotometer in the range of 400–700 nm using ddH_2_O as a blank [9].

#### 2.5.2. Detection of the Hard Protein Corona: Z-Potential and DLS Measurements

After the formation of the hard protein corona, the size of both types of AuNP was determined. In this step, only the AuNPs with the highest plasma concentrations (i.e., 55%) were analyzed to evaluate the size of the corona, i.e., the maximum thickness of the protein corona, on the surface of the nanoparticles.

#### 2.5.3. SDS-PAGE

Prior to SDS-PAGE, the total protein concentration of all samples was determined using the Bradford assay. For this purpose, 100 µL of reagent was mixed with 0–8 µL of standard or/and samples. Standard 1 was 1 mg/mL bovine serum albumin (BSA, Sigma-Aldrich, Prague, Czech Republic) and Standard 2 was the positive control and 100% BSA. Six different volumes were used for both standards and samples (0, 0.5, 1, 2, 4, 8 µL). Using the different concentrations of the standard (BSA), the standard curve was generated using a Tecan instrument at a wavelength of 595 nm, and the amounts of proteins in the nanoparticle dispersions were analyzed [35].

The hard protein corona was removed from the surface of the washed AuNP samples (after incubation with plasma) using 4X NuPAGE LDS Sample Buffer (Thermo Fisher Scientific, Prague, Czech Republic). Following that, 6.25 µL of the LDS sample buffer was mixed with 18.75 µL of each sample and incubated at 100 °C for 10 min. Samples were placed on ice for immediate cooling and then subjected to one-dimensional gel electrophoresis using NuPAGE 10% Bis-Tris Gel and a 1.0 mm × 10 well (Invitrogen, Thermo Fisher Scientific, Prague, Czech Republic) in the presence of 1X NuPAGE MES SDS running buffer (Thermo Fisher Scientific, Prague, Czech Republic). Two gels were run in an XCell SureLock™ Mini-Cell Tank (Thermo Fisher Scientific, Prague, Czech Republic) in the presence of Novex Sharp pre-stained protein standard (Thermo Fisher Scientific, Prague, Czech Republic) at 100 V, 150 mA, 100 W for 45 min. Finally, the gels were stained with SimplyBlue SafeStain (Thermo Fisher Scientific, Prague, Czech Republic) for 1 h with shaking. After washing the gels with ddH_2_O for 1 h, protein bands were visualized [36].

To detect the capping proteins (samples were not incubated with plasma), we ran two of the control samples (“cold” and “hot” AuNPs) in another experiment in the presence of 1X NuPAGE MOPS SDS running buffer (Thermo Fisher Scientific, Prague, Czech Republic) and 1X NuPAGE MES SDS running buffer, separately. We analyzed both buffers because there are some reports [37,38,39] that MES can complex with some metal ions in the sample, and we tried to see the differences between the buffers used. Two methods were used for sample preparation as well. The first was the same as described above (i.e., incubation of the samples with LDS at 100 °C for 10 min, referred to here as 100 °C), and in the second method, 6.25 µL of the LDS sample buffer was mixed with 18.75 µL of each sample and incubated at 110 °C for 15 min, after which the samples were immediately cooled on ice (referred to here as 110 °C). The sample sizes were analyzed using Thermo Scientific PageRuler Prestained Protein Ladder (Thermo Fisher Scientific, Prague, Czech Republic).

#### 2.5.4. LC-MS Analysis

Samples were mixed with 30 µL of 50 mM ammonium bicarbonate (Sigma-Aldrich, Prague, Czech Republic) and dithiothreitol (DTT, Sigma-Aldrich, Prague, Czech Republic) at a final concentration of 10 mM and incubated at 60 °C for 40 min. After cooling the samples to room temperature (RT), iodoacetamide (Sigma-Aldrich, Prague, Czech Republic) was added at a final concentration of 30 mM, and all samples were incubated in the dark for 30 min. To stop the alkylation reaction, DTT was added at a final concentration of 50 mM. Trypsin (Sigma-Aldrich, Prague, Czech Republic) at a concentration of 0.1 µg/mL was added, and samples were incubated overnight at 37 °C.

A liquid chromatography system (Agilent 1200 series, Agilent Technologies, Santa Clara, MA, USA) connected to a timsTOF Pro PASEF mass spectrometer with CaptiveSpray (Bruker Daltonics, Billerica, MA, USA) and operated in positive data-dependent mode was used to analyze the samples. Five microliters of each dispersion were injected by autosampler into the C18 trap column (UHPLC Fully Porous Polar C18 2.1 mm ID, Phenomenex). The flow rate was 20 µL/min, and after 5 min of injection, peptides were separated in the C18 column (Luna Omega 3 μm Polar C18 100 Å, 150 × 0.3 mm, Phenomenex) by a linear 35 min water-acetonitrile gradient (Sigma-Aldrich, Prague, Czech Republic) from 5% (*v*/*v*) to 35% (*v*/*v*) acetonitrile at a flow rate of 4 µL/min. Both the analytical and the trap columns were heated to 50 °C. For the timsTOF Pro settings, the parameters of the PASEF method for standard proteomics were used. Briefly, the intensity threshold was set to 1500, and the target intensity per individual PASEF precursor was set to 6000. The scan range was between 0.6 and 1.6 V s/cm^2^ with a ramp time of 100 ms, and the number of PASEF MS/MS scans was 10. Precursor ions were selected for fragmentation in an *m*/*z* range between 100 and 1700 with charge states ≥2+ and ≤6+, and active exclusion was activated for 0.4 min.

Proteomics data were processed using PEAKS Studio 10.0 software (Bioinformatics Solutions, Waterloo, ON, Canada). The parameters were as follows: The enzyme was trypsin (specific), with carbamidomethylation as fixed modification. The variable modifications were oxidation of methionine and acetylation of the N-terminus of the protein. The database was UniProt (all taxa, 11/2021). For protein corona, results were matched with the UniProt database for mammalian proteins, and for capping substance analysis, results were matched with the UniProt database for fungal proteins [40]. In addition, label-free quantification (LFQ) analysis was performed using the Perseus software suite, and volcano plots were obtained (http://www.perseusframework.org (accessed on 22 February 2022) [40].

## 3. Results and Discussion

### 3.1. Fungal Culturing and AuNP Production

Figure 1 shows the results obtained after incubation of the “cold” sample and after heating the “hot” sample for 5 min.

According to Figure 1, the color of the dispersions changed from yellow to dark red, indicating the formation of AuNPs. The negative control flask showed no color change.

### 3.2. Characterization of AuNPs

#### 3.2.1. Characterization of AuNPs Using Visible Spectrophotometry

Figure 2 shows the spectrophotometry results obtained for both (“hot” and “cold”) samples.

The spectrophotometry results showed that there were maximum absorption peaks between 500–550 nm for both types of AuNPs, indicating the formation of AuNPs in both samples. The maximum absorption peak for the “hot” AuNP dispersion was 528 nm, and for the “cold” AuNP sample, it was 541 nm (Figure 2).

The spectrophotometer results show that the AuNPs prepared at a higher temperature and in a shorter time have a maximum absorption peak at a lower wavelength, which is a first indication of a better size of AuNPs in the dispersion of the “hot” sample. As previously reported, the presence of a strong reducing agent in the mixture leads to a fast reduction rate with smaller nanoparticles, and here we have shown that by changing a single parameter (i.e., temperature), we can obtain smaller AuNPs. However, whether these nanoparticles will have the same properties as “cold” AuNPs must be investigated in the next steps.

#### 3.2.2. TEM Characterizations

Figure 3 shows the TEM images obtained. The results from EDS for both samples confirm the presence of the elemental Au in the samples (Figure 4).

The images from TEM confirm the presence of irregular AuNPs with a wide diameter range between 5–50 nm in the “cold” sample and spherical AuNPs with a diameter of 5–16 nm in the “hot” sample (Figure 3).

Figure 4 shows that in both the “cold” and “hot” samples, some Au peaks can be seen coming from the particles and not from the background of the grids, indicating the nature of the particles. From the results of TEM in this step, we can conclude that the “hot” AuNPs have smaller sizes, as we hypothesized in the spectrophotometry part.

#### 3.2.3. Characterization of AuNPs: Z-Potential and DLS Measurements

Size analysis of two different “cold” and “hot” AuNP samples confirmed the results of TEM, and the average size of the “hot” sample was smaller than that of the “cold” sample (13.1 nm ± 2.1 vs. 37.9 nm ± 5.7, respectively). When the samples were analyzed for zeta potential, the average zeta potential was slightly higher for the “cold” sample than the “hot” sample (−39.4 mV ± 5.13 vs. −35.8 mV ± 4.15, respectively). These results are shown in Table 1.

#### 3.2.4. FTIR Characterization

FTIR results for both ”cold” and “hot” samples are represented in Figure 5.

According to Figure 5, both of the “cold” and “hot” AuNPs have a similar spectral shape in the diagnostic region (3000–1500 cm^−1^). According to Figure 5, the FTIR spectrum of the “hot” AuNPs has a broad and intense peak at 3315 cm^−1^ corresponding to the O-H stretching vibration and N-H stretching of the secondary amines. A weak shoulder at about 3070 cm^−1^ corresponds to the C-H stretching of alkene (=C-H). The peak at 2928 cm^−1^ with its nearby overtones represents the C^sp3^ -H stretching for symmetric and asymmetric stretching vibrations of alkanes. For “cold” AuNPs, the FTIR spectrum exhibits -OH and -NH groups, giving a broad peak of their stretching at 3297 cm^−1^. In addition, there are 2 shoulders at about 3190 cm^−1^ and 3072 cm^−1^. The presence of these two peaks together with the peak at about 3300 cm^−1^, which overlaps with the intense OH peak, indicates the presence of the primary NH_2_ group: symmetric and asymmetric N-H stretching vibrations with their overtones at 3072 cm^−1^ and 3190 cm^−1^. The N-H-related peaks are broadened due to hydrogen bonding. Intense peaks at 1655 cm^−1^ (“hot”) and at 1650 cm^−1^ (“cold”) are consistent with C=O stretching of amides. It is broader for “cold” AuNPs than for “hot” AuNPs, indicating their hydrogen bonding. Both the “cold” and “hot” AuNP peaks at 1655, 1547, and 1452 cm^−1^ most likely originate from amide vibrations known for the amide I, amide II, and amide II of the proteins, respectively (Figure 5).

Based on the obtained data from control AuNPs, among biomolecules, only proteins and peptides can exhibit such IR profiles.

### 3.3. Detection of the Hard Protein Corona

To analyze the hard protein corona on the surface of washed AuNPs, the following methods were used:

#### 3.3.1. Detection of the Hard Protein Corona Using Visible Spectrophotometry

All samples were analyzed with a Nanodrop spectrophotometer after the formation of the protein corona. Figure 6 shows the obtained results.

The results show that after protein corona formation the maximum absorbance peak of the “cold” and “hot” samples changed slightly and that the complete formation of the protein corona made the AuNPs in the “cold” sample unstable, because it seems that the peaks were not sharp, and their slopes were wider than those of the control AuNPs. Although both the “cold” and “hot” samples had wider slopes after protein corona formation in contrast to the controls, the “hot” AuNPs were more stable and resistant to agglomeration after protein corona formation than the “cold” AuNPs.

Since the optical absorption of proteins is at 280 nm [41], the presence of absorption in all spectra between 500–500 nm is due to the presents of particles in the nano-dimensions.

#### 3.3.2. Detection of the Hard Protein Corona: Z-Potential and DLS Measurements

The size and zeta potential of the two types of “cold” and “hot” AuNPs after incubation with 55% plasma, the highest concentration used in this study, were checked with the Zetasizer to show whether the formation of the protein corona had an effect on the size and surface charge of the bothAuNP samples. DLS analysis was used to assess the thickness of the protein corona. Table 1 and Figure 7 show the obtained results for all of the samples. Samples were checked for size distribution by number (Table 1) and by intensity (Figure 7).

As the results show (Table 1), the AuNPs in the “cold” sample had two distinguishable peaks, indicating that the formation of the protein corona made this type of nanoparticle polydisperse, while this was not the case for the “hot” AuNP sample, suggesting that this type of nanoparticle is more stable than the “cold” type. These results confirm the spectrophotometry results for the same nanoparticles.

Table 1 shows that after the formation of the protein corona on the surface of the two types of “cold” and “hot” AuNPs, the size of AuNPs increased (37.9 nm vs. 109.3 nm and 13.1 nm vs. 86.7 nm, respectively), and the zeta potential decreased (−39.4 mV vs. −37.7 mV and −35.8 mV vs. −30.5 mV, respectively). This decrease was not so strong that the AuNPs would become unstable and agglomerated, but it is obvious that a thick protein corona was formed on the surface of the AuNPs which is larger than the diameter of the AuNPs.

According to Figure 7, both types of AuNPs before the formation of the protein corona had one peak of size distribution by intensity (Figure 7A), but after the formation of the protein corona, the nanoparticles were polydispersed and had two peaks (Figure 7B), which support the Table 1 results. According to these findings of distribution by intensity, both samples have two populations of different sizes of particles, but after the math recalculations into a number distribution, the second population became indistinguishable for the “hot” sample.

### 3.4. SDS-PAGE and Detected Protein Coronas

First, a Bradford assay was performed to determine the total protein concentration of all samples. We do not present the results of this assay here because the AuNPs in all wells tested showed some absorbance that interfered with the wavelength used, and the results obtained were not reliable.

The results from SDS-PAGE confirmed the presence of a hard protein corona around both “hot” and “cold” AuNPs. Figure 8 shows the obtained results.

As the results in Figure 8 (the results from using MES running buffer) show, the highest plasma concentration resulted in the largest amounts of protein corona on the surface of the AuNPs. To better assess the results, a protein standard pre-stained with Novex Sharp was loaded into each panel in Figure 8. Most proteins were between 60–80 kDa and 10–3.5 kDa. The LDS sample buffer was used as a control to ensure that the buffer was not contaminated by proteins. In addition, both types of control AuNPs were analyzed, and the results showed that there was no protein band in either sample and that there was a protein band in the 60–80 kDa range that could be an impurity from the next well (Figure 8C,D), which accounted for 55% of the plasma. Although the FTIR results showed the presence of proteins or peptides on the surface of the control AuNP samples as capping agents, the amount of these proteins is so small that it cannot be detected by SDS-PAGE. In this study, both AuNPs were boiled in the presence of LDS for 10 min as a control and then used for SDS-PAGE analysis, so the detachment of the capping protein(s) from the AuNPs was achieved by this method. As shown in Figure 8B,C, the AuNPs were trapped at the top of the gel, and there were no protein bands in the two control “cold” and “hot” samples. The presence of nanoparticles at the top of the polyacrylamide gel has been reported previously, although the nanoparticles had a negative zeta potential and were expected to migrate down the gel in the presence of LDS, which did not occur. In contrast to previous studies that showed that the biologically produced AuNPs had proteinaceous caps [23], here we assume that if there are some proteins on the surface of the nanoparticles, their amounts are too small to be detected by the SDS-PAGE method. This suggests that *F. oxysporum* may use other types of capping agents. It has been reported that two proteins produced by *F. oxysporum* functioned as AuNP caps [23], which was not the case here. This difference could be due to the extensive washing process used in our study compared with Zhang et al. [23]. The other possibility is that only a very small fraction of the proteins are present as capping agents on the surface of the nanoparticles, as we detected their presence by FTIR. The use of LC-MS will clarify this issue.

Figure 9 shows the results of the “cold” and “hot” AuNP samples in MES and MOPS running buffers.

As the results show (see Figure 9), there were no protein bands under the different sample preparation conditions and in two different running buffers. Therefore, in our experiment the use of SDS-PAGE could not reveal the capping agents.

### 3.5. LC-MS Analysis and Proteomics Data

As mentioned in the Materials and Methods section, all samples—two “cold” and “hot” AuNP samples, each with three different plasma concentrations (10, 25, and 55%) and two AuNP control samples without plasma—were prepared in triplicate and analyzed by LC-MS. The results obtained were first analyzed using PEAKS Studio 10.0 and then using Perseus software. Table 2 summarizes the most abundant proteins with the highest coverage that were present in all replicates of AuNP plasma samples (AuNPs with 10, 25 and 55% plasma).

According to Table 2, it is evident that the surfaces of AuNPs were mainly covered by subunits of fibrinogen, hemoglobin, keratin, actin, ribosomal proteins, and some enzymes such as biliverdin reductase and glutathione peroxidase. All samples had a protein corona, and this table shows the common proteins in AuNP plasma samples and not in AuNP control samples. It seems that these major proteins are not the ones that help the immune system to find and mark foreign molecules. On the other hand, they might help the AuNPs escape recognition by phagocytes by shielding the nanoparticles.

It is important to note that even though we used plasma and not whole blood, there was still some hemoglobin present due to remnants of red blood cells. We believe a similar situation will occur after application of the AuNPs in vivo. Table 3 shows the results for the control AuNPs.

Table 3 shows that the proteins had low coverage percentages. Therefore, we can conclude that it might be the case that some amino acids or small peptides and almost no proteins are present on the surface of AuNPs that can act as capping agents. These small amounts of proteins are better covered in the “cold” AuNP sample than in the “hot” one, suggesting that the use of a higher incubation time (slow process) gives the proteins from the supernatant of the fungal culture some time to adsorb onto the surface of AuNPs.

In our previous studies, we tested *F. oxysporum* for its ability to produce AuNPs, and we found that fungi could produce AuNPs with an average size of 20 nm. The obtained AuNPs were washed, and TEM images proved that organic materials were removed from the surface of the AuNPs during the washing process [8]. Proteins represent one of the main organic components within the microbial culture. According to our current results, we conclude that washing the nanoparticles eliminated most of those proteins. The remaining amounts of microbial proteins were too low to be detected by a sensitive approach such as LC-MS. We recommend evaluating the organic matter content of the biologically produced AuNPs in future. Perseus software was used to determine the differences in proteins in the coronas of each group of AuNPs. For this purpose, AuNPs with different plasma content levels were divided into 3 groups: (1) “cold” and “hot” AuNPs with 10% plasma content, (2) “cold” and “hot” AuNPs with 25% plasma content, and (3) “cold” and “hot” AuNPs with 55% plasma content. The obtained volcano curves are shown in Figure 10.

Figure 10 shows that the type of protein corona differs between “hot” and “cold” AuNPs in each group. Thus, it appears that the protein corona on the surface of the biologically produced AuNPs differs according to the capping agents of each “hot” and “cold” AuNP sample and according to the concentration of the plasma. Further studies are needed to understand the surface composition of the biologically AuNPs produced by *F. oxysporum*.

There are some reports on the use of dead biomass of microorganisms for non-enzymatic bio-reduction processes [42,43,44,45]. In this study, we used extracellularly secreted biomolecules that can act as reducing and capping agents at high temperatures (i.e., the method used for the “hot” AuNPs sample). From the results and the differences between the “hot” and the “cold” AuNPs, we concluded that although the AuNPs were produced by the same organism with the same secreted substances, their capping and reducing agents were probably different, which must be further investigated in the future.

In our previous research, as well as in another research study [46], it was shown that in the case of nanodiamonds (ND), the saturation of the surface of the NDs with transferrin before their exposure to plasma prevents the formation of a protein corona

## 4. Conclusions

Despite several studies on the presence of proteins on the surface of biologically produced nanoparticles that served as capping agents, the current study showed that the capping agents of AuNPs produced by *F. oxysporum* at two different temperatures were not proteins. It is possible that proteins were present in amounts too low to detect. Small peptides, amino acids, or other types of capping agents are conceivable and should be defined in the future. Our research showed that produced AuNPs are stable beyond their chemical composition even after protein corona formation. In addition, we found that AuNPs prepared at higher temperature and in shorter time exhibited better properties in terms of size and zeta potential both before and after protein corona formation, which make them a good candidate for in vivo analysis. The protein coronas of both “cold” and “hot” AuNPs consisted of proteins that protected the NPs from phagocyte recognition, which supports their use for in vivo drug delivery.

## Figures and Tables

**Figure 1 materials-15-04615-f001:**
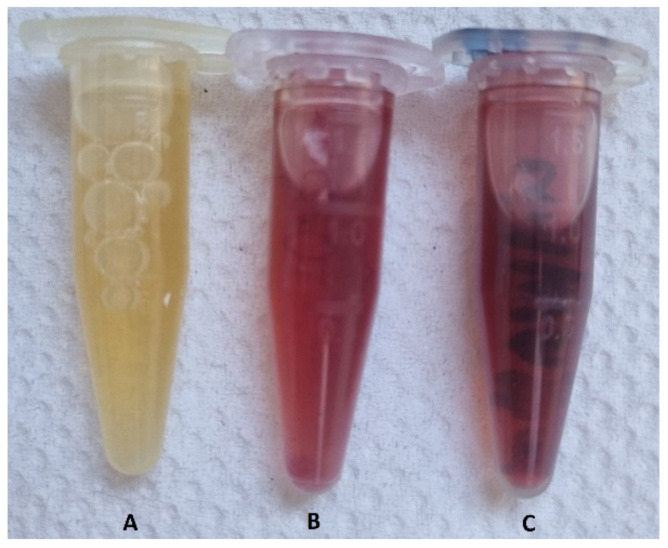
The changed color of the dispersions after incubation at two different “cold” and “hot” temperatures. (**A**) the negative control sample, (**B**) the sample incubated at 37 °C (i.e., “cold”), and (**C**) the sample heated at 80 °C (i.e., “hot”).

**Figure 2 materials-15-04615-f002:**
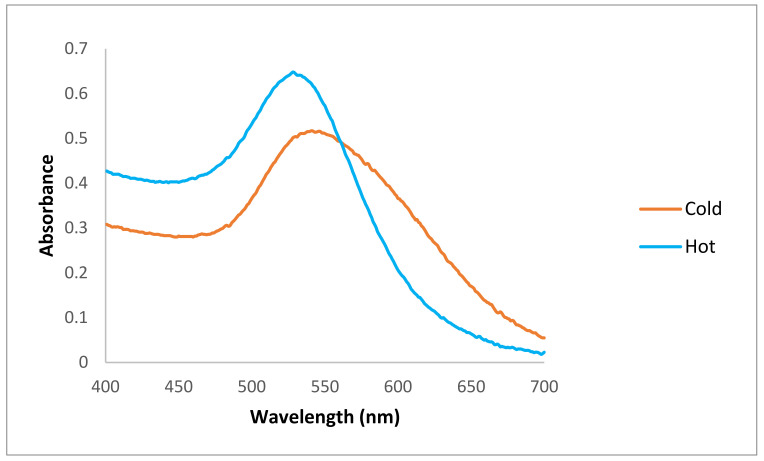
Spectrophotometry results of produced AuNPs. The maximum absorption peak for the “hot” AuNP dispersion was 528 nm (the blue line) and for the “cold” AuNP sample was 541 nm (the orange line).

**Figure 3 materials-15-04615-f003:**
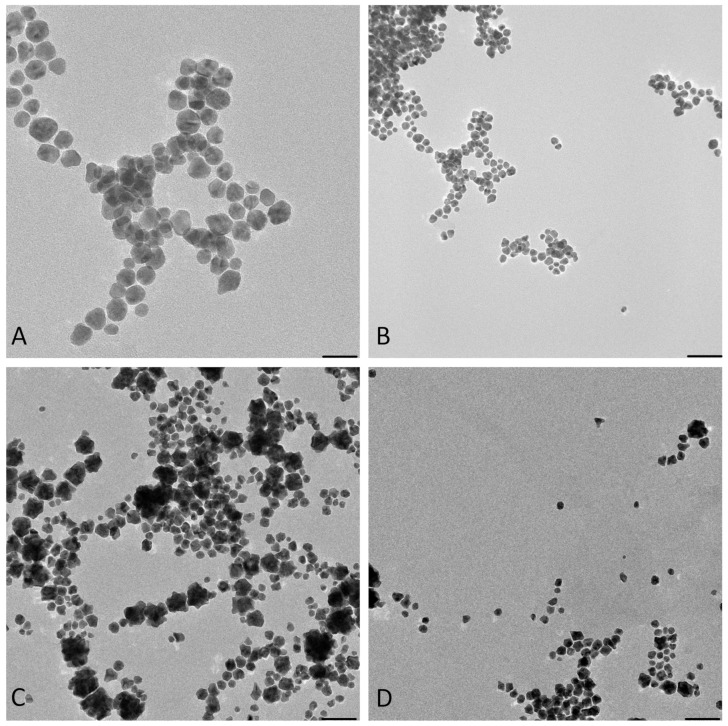
TEM-obtained images of “cold” and “hot” AuNP samples. (**A**,**B**): obtained images from the “hot” sample. (**C**,**D**): obtained images from the “cold” sample. (Scale bars for **B**–**D** = 50 nm and for **A** = 20 nm). (**E**): Size distribution histograms of “cold” and “hot” AuNP samples.

**Figure 4 materials-15-04615-f004:**
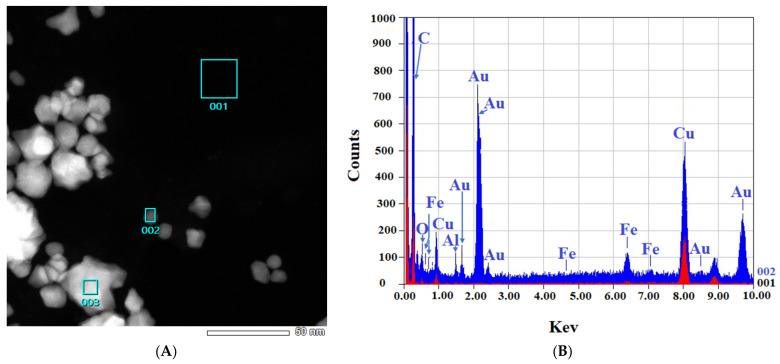
EDS-obtained images of “cold” and “hot” AuNP samples. (**A**,**B**): Obtained images from the “cold” sample. (**C**,**D**): Obtained images from the “hot” sample. (Scale bars = 50 nm). The spectra for the background (red) and AuNPs (blue) reveal the differences.

**Figure 5 materials-15-04615-f005:**
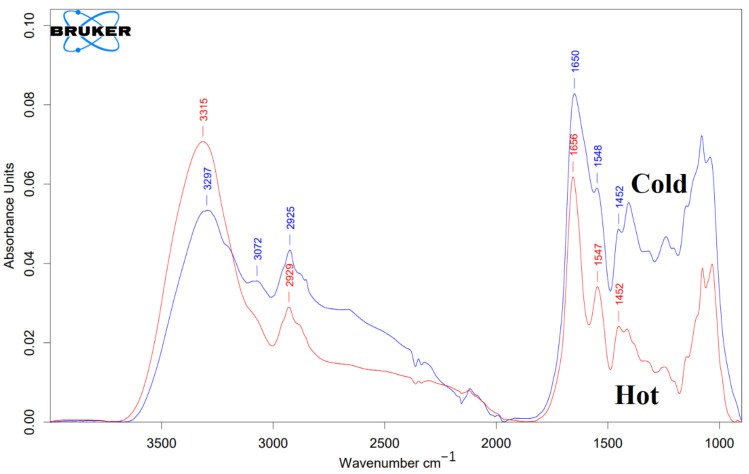
FTIR characterization of both “cold” and “hot” AuNPs. The blue line represents “cold” AuNPs, and the red line represents “hot” AuNPs. As mentioned in the text, on the surface of the control AuNPs (without protein corona), there are peptides or proteins that act as capping agents.

**Figure 6 materials-15-04615-f006:**
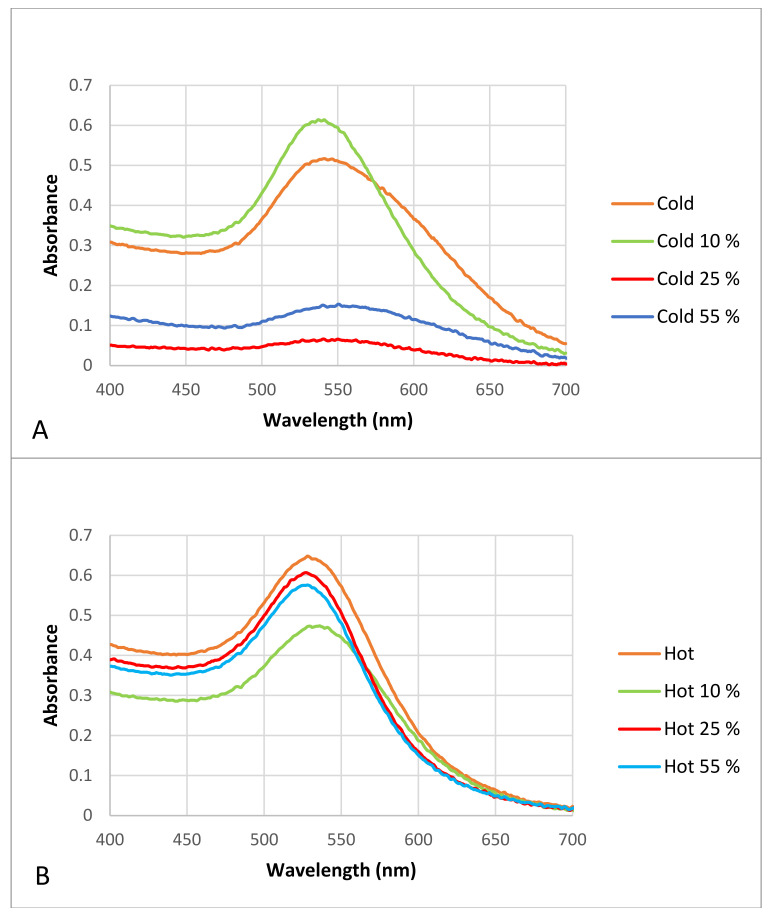
Visible spectrophotometry of the “cold” and “hot” AuNP samples after incubation with different plasma concentrations. (**A**) “cold” AuNPs as control in contrast to AuNPs with different proportions of plasma (i.e., protein corona), (**B**) “hot” AuNPs as control in contrast to AuNPs with different proportions of plasma (i.e., protein corona).

**Figure 7 materials-15-04615-f007:**
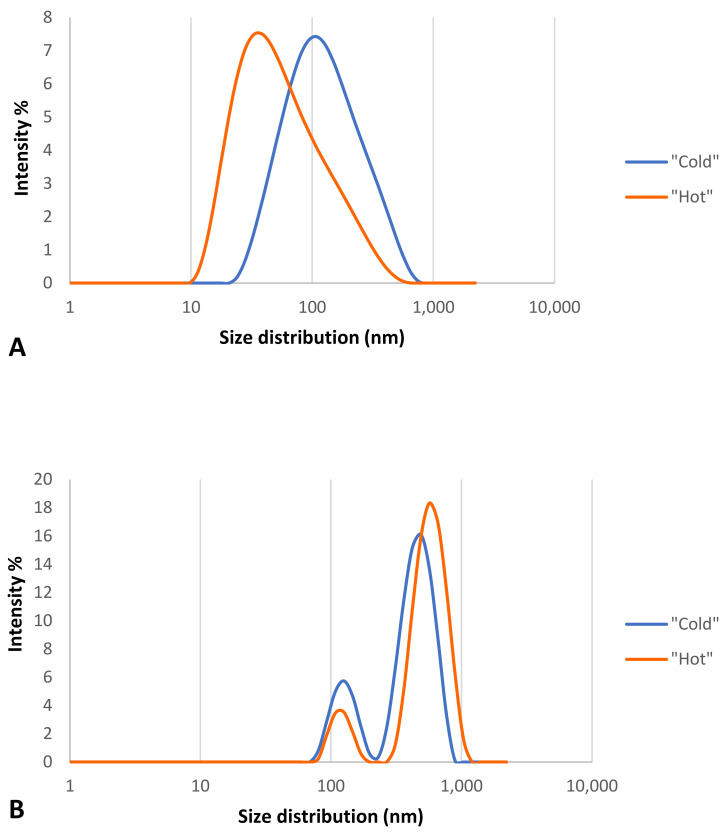
Size distribution analysis graph for “cold” and “hot” AuNPs by intensity. (**A**) before corona formation and (**B**) after corona formation.

**Figure 8 materials-15-04615-f008:**
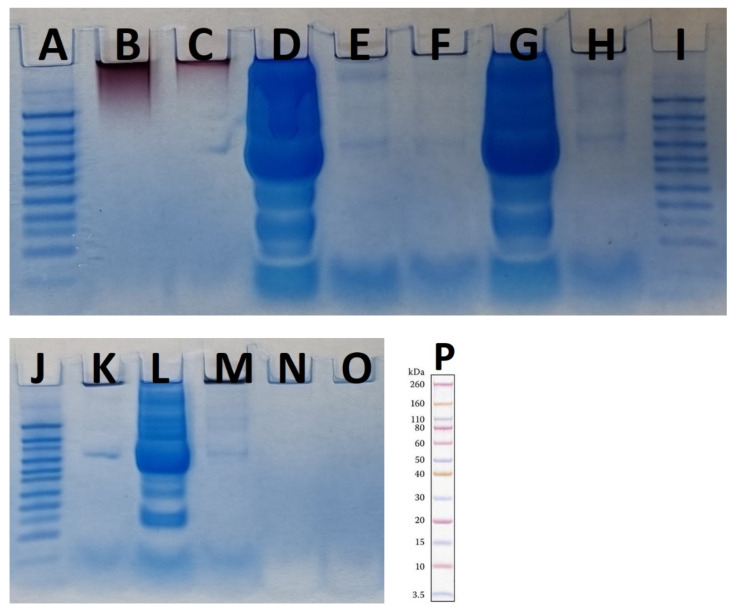
Two SDS-PAGE gels for different concentrations of plasma as controls and AuNPs with different concentrations of plasma as tests in the presence of MES running buffer. (**A**,**I**,**J**,**P**) are Novex Sharp pre-stained protein standard. (**B**) “hot” AuNPs, (**C**) “cold” AuNPs, (**D**) plasma 55%, (**E**) “hot” AuNPs—plasma 55%, (**F**) “cold” AuNPs—plasma 55%, (**G**) plasma 25%, (**H**) “hot” AuNPs—plasma 25%, (**K**) “cold” AuNPs—plasma 25%, (**L**) plasma 10%, (**M**) “hot” AuNPs—plasma 10%, (**N**) “cold” AuNPs—plasma 10%, and (**O**) LDS sample buffer as control.

**Figure 9 materials-15-04615-f009:**
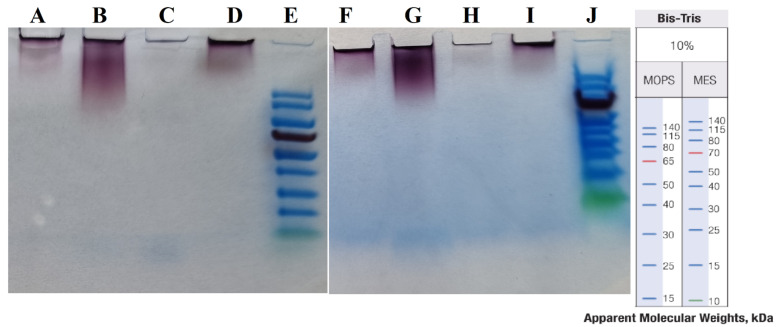
SDS-PAGE in two different buffers. (**A**–**E**) in MOPS and (**F**–**J**) in MES buffers. (**E**,**J**) are Thermo Scientific PageRuler Prestained Protein Ladders, with size shown on right. (**A**,**F**) “cold” AuNPs at 100 °C, (**B**,**G**) “hot” AuNPs at 100 °C, (**C**,**H**) “cold” AuNPs at 110 °C, and (**D**,**I**) “hot” AuNPs at 110 °C.

**Figure 10 materials-15-04615-f010:**
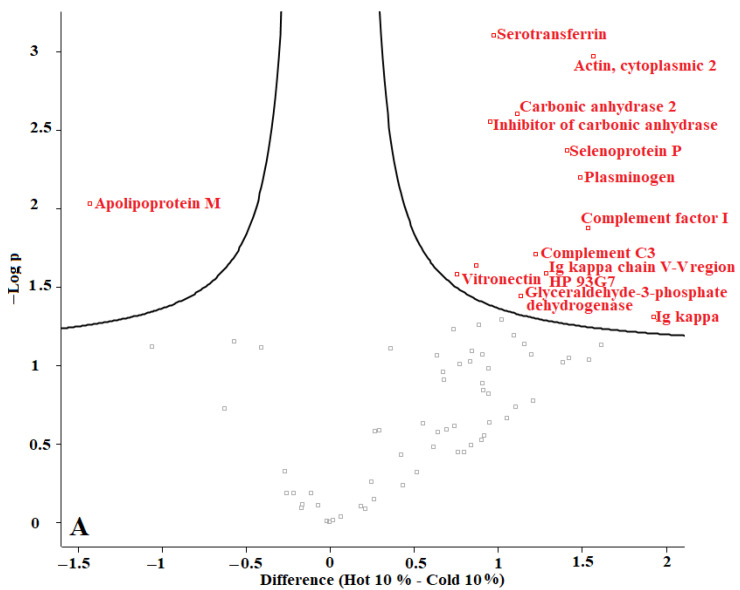
Volcano plots generated with Perseus software for 3 different groups of AuNPs. (**A**) differences in protein coronas of “cold” and “hot” AuNPs with 10% plasma content, (**B**) “cold” and “hot” AuNPs with 25% plasma content, and (**C**) “cold” and “hot” AuNPs with 55% plasma content. According to the horizontal axes, from 0 to the left is the “hot” protein corona, and from 0 to the right is the “cold” protein corona, showing their differences.

**Table 1 materials-15-04615-t001:** The obtained size distribution by number (nm) and zeta potential (mV) of two different “cold” and “hot” AuNP samples in 3 (size) to 5 (zeta potential) replicates. Size and zeta potential of AuNP samples after incubation with 55% plasma in different replicates are shown and compared here. NA stands for not applicable.

**AuNP Samples Before Protein Corona Formation**	**Mean**	**SD**
Size of “cold” (nm)	37.9	5.70
Size of “hot” (nm)	13.1	2.10
Zeta potential (mV) of “cold”	−39.4	0.65
Zeta potential (mV) of “hot”	−35.8	1.17
**AuNP Samples After Protein Corona Formation**	**Mean**	**SD**
Size of “cold” peak I (nm)	109.3	23.12
peak II (nm)	267.0	0
Size of “hot” peak I (nm)	86.7	25.04
peak II (nm)	NA	NA
Zeta potential (mV) of “cold”	−37.7	5.13
Zeta potential (mV) of “hot”	−30.5	4.15

**Table 2 materials-15-04615-t002:** The results of LC-MS analysis using the software PEAKS Studio 10.0. Here are listed the most abundant proteins with the highest coverage present in all replicates of AuNP plasma samples (AuNPs with 10%, 25% and 55% plasma). As mentioned earlier, the results obtained were compared against the UniProt mammalian protein database.

Protein ID	Description	Accession	Total Coverage (%)	Peptides	Unique	Avg. Mass
28	Fibrinogen beta chain OS = Rattus norvegicus OX = 10,116 GN = Fgb PE = 1 SV = 4	P14480|FIBB_RAT	42	24	15	54,235
556	Hemoglobin subunit alpha OS = Otospermophilus beecheyi OX = 34,862 PE = 1 SV = 1	B3EWC9|HBA_OTOBE	52	8	4	15,023
92	Fibrinogen beta chain OS = Cavia porcellus OX = 10,141 GN = FGB PE = 4 SV = 2	tr|H0VD80|H0VD80_CAVPO	28	16	6	54,105
258	Fibrinogen beta chain OS = Oryctolagus cuniculus OX = 9986 GN = FGB PE = 4 SV = 1	tr|A0A5F9D3P7|A0A5F9D3P7_RABIT	24	13	4	53,716
278	Fibrinogen beta chain OS = Oryctolagus cuniculus OX = 9986 GN = FGB PE = 4 SV = 3	tr|G1T0W8|G1T0W8_RABIT	23	13	4	56,507
818	Hemoglobin subunit alpha OS = Blarina brevicauda OX = 9387 PE = 1 SV = 1	B3EWE1|HBA_BLABR	43	6	4	14,995
63	Keratin 5 OS = Oryctolagus cuniculus OX = 9986 GN = KRT5 PE = 3 SV = 1	tr|A0A5F9CNQ8|A0A5F9CNQ8_RABIT	32	26	6	60,612
59	Keratin 10 OS = Oryctolagus cuniculus OX = 9986 GN = KRT10 PE = 3 SV = 1	tr|A0A5F9D8K0|A0A5F9D8K0_RABIT	26	20	0	59,920
60	Keratin 10 OS = Oryctolagus cuniculus OX = 9986 GN = KRT10 PE = 3 SV = 2	tr|G1T1V0|G1T1V0_RABIT	26	20	0	56,342
36	Actin gamma 1 OS = Cavia porcellus OX = 10,141 GN = ACTG1 PE = 3 SV = 1	tr|A0A286XYY5|A0A286XYY5_CAVPO	55	18	7	41,793
95	Keratin 10 OS = Myotis lucifugus OX = 59,463 GN = KRT10 PE = 3 SV = 1	tr|G1P6A9|G1P6A9_MYOLU	20	17	0	58,383
701	Hemoglobin subunit alpha OS = Microtus pennsylvanicus OX = 10,058 PE = 1 SV = 1	B3EWE3|HBA_MICPE	45	8	3	15,073
120	IF rod domain-containing protein OS = Cavia porcellus OX = 10,141 PE = 3 SV = 1	tr|A0A286XNZ7|A0A286XNZ7_CAVPO	23	17	2	58,626
363	Hemoglobin subunit beta OS = Microtus pennsylvanicus OX = 10,058 PE = 1 SV = 1	B3EWE4|HBB_MICPE	62	9	6	15,677
585	Hemoglobin subunit alpha OS = Peromyscus californicus OX = 42,520 PE = 1 SV = 1	B3EWD5|HBA_PERCA	68	8	2	14,869
1141	Hemoglobin subunit alpha OS = Tamiasciurus hudsonicus OX = 10,009 PE = 1 SV = 1	B3EWD7|HBA_TAMHU	38	6	2	14,986
664	Globin A1 OS = Myotis lucifugus OX = 59,463 GN = GLNA2 PE = 3 SV = 1	tr|G1QEL0|G1QEL0_MYOLU	47	8	4	15,885
437	Hemoglobin subunit beta OS = Tamiasciurus hudsonicus OX = 10,009 PE = 1 SV = 1	B3EWD8|HBB_TAMHU	50	7	3	15,855
676	Hemoglobin subunit beta OS = Otospermophilus beecheyi OX = 34,862 PE = 1 SV = 1	B3EWD0|HBB_OTOBE	31	6	3	15,825
767	Hemoglobin subunit alpha OS = Sciurus carolinensis OX = 30,640 PE = 1 SV = 1	B3EWD1|HBA_SCICA	57	7	3	15,073
849	Hemoglobin subunit alpha OS = Peromyscus crinitus OX = 144,753 PE = 1 SV = 1	B3EWD3|HBA_PERCR	48	6	2	14,986
426	Hemoglobin subunit beta OS = Peromyscus crinitus OX = 144,753 PE = 1 SV = 1	B3EWD4|HBB_PERCR	47	6	1	15,807
727	Biliverdin reductase B OS = Cavia porcellus OX = 10,141 GN = BLVRB PE = 4 SV = 1	tr|H0UVS9|H0UVS9_CAVPO	36	5	3	22,096
778	Biliverdin reductase B OS = Myotis lucifugus OX = 59,463 GN = BLVRB PE = 4 SV = 1	tr|G1P4F0|G1P4F0_MYOLU	25	4	2	21,982
660	Hemoglobin subunit alpha OS = Tamias merriami OX = 123,787 PE = 1 SV = 1	B3EWC7|HBA_TAMMR	62	7	4	15,061
774	Hemoglobin subunit beta OS = Blarina brevicauda OX = 9387 PE = 1 SV = 1	B3EWE2|HBB_BLABR	27	4	1	15,795
730	Hemoglobin subunit beta OS = Tamias striatus OX = 45,474 PE = 1 SV = 1	B3EWE0|HBB_TAMST	27	4	1	15,869
795	Hemoglobin subunit alpha OS = Vicugna pacos OX = 30,538 GN = HBA PE = 1 SV = 1	P67816|HBA_VICPA	38	4	2	15,126
896	Hemoglobin subunit alpha OS = Lama vicugna OX = 9843 GN = HBA PE = 1 SV = 1	P07425|HBA_LAMVI	38	4	2	15,142
880	Hemoglobin subunit alpha OS = Tamias striatus OX = 45,474 PE = 1 SV = 1	B3EWD9|HBA_TAMST	37	4	1	15,154
839	Glutathione peroxidase OS = Oryctolagus cuniculus OX = 9986 PE = 3 SV = 1	tr|A0A5F9CNR1|A0A5F9CNR1_RABIT	21	4	2	23,298
1065	Ubiquitin-like domain-containing protein OS = Oryctolagus cuniculus OX = 9986 PE = 4 SV = 1	tr|A0A5F9CP41|A0A5F9CP41_RABIT	44	3	3	8694
1067	60S ribosomal protein L40 OS = Takifugu rubripes OX = 31,033 GN = uba52 PE = 3 SV = 1	tr|H2SBM2|H2SBM2_TAKRU	27	3	3	14,745
1068	60S ribosomal protein L40 OS = Cavia porcellus OX = 10,141 GN = Uba52 PE = 3 SV = 1	tr|A0A286XB24|A0A286XB24_CAVPO	27	3	3	14,728

**Table 3 materials-15-04615-t003:** Results of “hot” and “cold” AuNPs without incubation with plasma, which were used as controls. As mentioned above, results were compared against the UniProt fungi protein database.

Protein ID	Description	Accession	Coverage (%)	Peptides	Unique	Avg. Mass
**HHot**						
1383	Ubiquitin OS = Encephalitozoon cuniculi (strain GB-M1) OX = 284,813 GN = ECU02_0740i PE = 1 SV = 1	Q8SWD4|UBIQ_ENCCU	12	1	1	8714
**Cold**						
1389	Ubiquitin-60S ribosomal protein L40 OS = Neurospora crassa (strain ATCC 24,698/74-OR23-1A/CBS 708.71/DSM 1257/FGSC 987) OX = 367,110 GN = crp-79 PE = 1 SV = 2	P0C224|RL40_NEUCR	20	2	2	14,637
1386	Ubiquitin-60S ribosomal protein L40 OS = Schizosaccharomyces pombe (strain 972/ATCC 24843) OX = 284,812 GN = uep1 PE = 1 SV = 1	P0CH07|RL402_SCHPO	20	2	2	14,595
1387	Ubiquitin-60S ribosomal protein L40 OS = Saccharomyces cerevisiae (strain ATCC 204508/S288c) OX = 559,292 GN = RPL40A PE = 1 SV = 1	P0CH08|RL40A_YEAST	20	2	2	14,554
1388	Ubiquitin-60S ribosomal protein L40 OS = Saccharomyces cerevisiae (strain ATCC 204508/S288c) OX = 559,292 GN = RPL40B PE = 1 SV = 1	P0CH09|RL40B_YEAST	20	2	2	14,554
1390	Ubiquitin-60S ribosomal protein L40 OS = Schizosaccharomyces pombe (strain 972/ATCC 24843) OX = 284,812 GN = ubi1 PE = 1 SV = 1	P0CH06|RL401_SCHPO	20	2	2	14,595
1391	Ubiquitin-60S ribosomal protein L40 OS = Cryptococcus neoformans var. neoformans serotype D (strain JEC21/ATCC MYA-565) OX = 214,684 GN = UBI1 PE = 1 SV = 2	P40909|RL40_CRYNJ	19	2	2	14,653
1397	Ubiquitin-40S ribosomal protein S27b OS = Schizosaccharomyces pombe (strain 972/ATCC 24843) OX = 284,812 GN = ubi5 PE = 1 SV = 2	P0C8R3|RS27B_SCHPO	17	2	2	17,215
1393	Ubiquitin-40S ribosomal protein S27a OS = Schizosaccharomyces pombe (strain 972/ATCC 24843) OX = 284,812 GN = ubi3 PE = 1 SV = 2	P0C016|RS27A_SCHPO	17	2	2	17,258
1394	Ubiquitin-40S ribosomal protein S31 OS = Saccharomyces cerevisiae (strain ATCC 204508/S288c) OX = 559,292 GN = RPS31 PE = 1 SV = 3	P05759|RS31_YEAST	16	2	2	17,216
1395	Polyubiquitin OS = Candida albicans OX = 5476 GN = UBI1 PE = 1 SV = 1	P0CG73|UBI1P_CANAX	11	2	2	25,755

## Data Availability

Data available on request. The data from LC-MS analysis are not publicly available due to non-existing database. The data presented in this study are available on request from the corresponding author.

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
