# Peer review of "Investigation of Protein Corona Formed around Biologically Produced Gold Nanoparticles"

_materials, 2022, doi:10.3390/ma15134615_

Round 1
Reviewer 1 Report
Title: Evaluation the protein corona around the biologically produced
gold nanoparticles
Authors: Parastoo Pourali et al
Comments
The surface coating of any NPs, including AuNPs, with protein corona (PC) is indeed a prominent and currently unresolved topic. The binding of proteins to the nano-surface is a complex, and in practice not always predictable process.
The authors of this work investigate the protein corona around gold nanoparticles synthesized by green method.
I find the manuscript interesting and valuable. However, this paper needs some more work in order to meet the technical content criteria, as I explain below.
Specific comments:
· There are some technical details missing such as size distribution histograms for the AuNPs.
· (Title) Evaluation in title may not be appropriate, investigation may be better description of this work.
· (Abstract) This is a very confusing sentence: “ In contrast to the control samples, which contained almost no protein(s) and only small peptides/amino acids as capping agents, analysis of the AuNPs with hard protein corona showed that some proteins were present.” Should control samples be on their own as said before? What does “some proteins” mean?
· (intro) “Nanoparticles are particles that are smaller than 100 nm “I disagree with this statement. The term is sometimes used for larger particles, up to 500 nm. We have been able to culture skin cells with 200-300 nm nanoparticles.
· Suggestion to replace: “low” and “high” with “cold” and “warm” or similar
· Line 75: “linkers” could be changed to “binders”.
· Line 137 please specify mouse type.
· Line 156 If absorption only “may change…” than this technique is not suitable for PC detection.
· In all Results sections any figure with the results should be first introduced, then results should explained and then some conclusions can be drawn. It seems the opposite order in this manuscript has been applied.
· Figure 3: Higher level of agglomeration in “low’ sample could be discussed.
· Figure 4: Font size needs to be enlarged in B and D. This figure is hardly discussed in the text, may go to supplementary material.
· Line 275 “have a better size” needs to be rephrased.
· Line 278: “the size of the "high" sample was smaller than that of the "low" sample (37.9 278 nm vs. 13.1 nm, respectively” Did you mean mean/average size?
· Section: 3.8. Visible spectrophotometry. More discussion is needed for flattening the absorption. It is well known that protein absorption is quite flat after 400 nm.
· Line 330: “whether the formation of the protein corona had an effect on the nature of the AuNPs.” Nature of AuNPs? Maybe you think properties and please specify which one.
· Table 1 should contain units.
· Line 478: “made them a good candidate for in vivo analysis.” Made should be replaced with make
In conclusion, I find the results presented in this manuscript to merit publication in Materials, after substantial revision.
Author Response
Dear Reviewer,
Thank you for your stimulating comments and suggestions. We have added your inputs to the text and I believe it significantly improved quality of the manuscript.
Our corrections are listed below in detail.
Best regards,
Veronika Benson
Comments
- There are some technical details missing such as size distribution histograms for the AuNPs.
Yes, we have added the histograms.
- (Title) Evaluation in title may not be appropriate; investigation may be better description of this work.
Yes, we changed the title.
- (Abstract) This is a very confusing sentence: “ In contrast to the control samples, which contained almost no protein(s) and only small peptides/amino acids as capping agents, analysis of the AuNPs with hard protein corona showed that some proteins were present.” Should control samples be on their own as said before? What does “some proteins” mean?
We have edited the sentence as suggested.
- (intro) “Nanoparticles are particles that are smaller than 100 nm “I disagree with this statement. The term is sometimes used for larger particles, up to 500 nm. We have been able to culture skin cells with 200-300 nm nanoparticles.
We have deleted this part.
- Suggestion to replace: “low” and “high” with “cold” and “warm” or similar.
Yes, we replaced “low” and “high” with “cold“and “hot“.
6- Line 75: “linkers” could be changed to “binders”.
Yes, we replaced that.
7- Line 137 please specify mouse type.
We added the mouse strain.
8- Line 156 If absorption only “may change…” than this technique is not suitable for PC detection.
Yes, we have corrected the sentence.
9- In all Results sections any figure with the results should be first introduced, then results should explained and then some conclusions can be drawn. It seems the opposite order in this manuscript has been applied.
Yes, we have changed the order according to the respected reviewer’s comment.
10- Figure 3: Higher level of agglomeration in “low’ sample could be discussed.
It seems that the AuNPs in “low” sample were agglomerated but after zooming we can detect that the AuNPs were separated even in the close contact. Just their sizes were bigger and not uniform than “high” sample.
11- Figure 4: Font size needs to be enlarged in B and D. This figure is hardly discussed in the text, may go to supplementary material.
Yes, we have inserted the original files.
12- Line 275 “have a better size” needs to be rephrased.
Yes, we corrected that to “smaller size”.
13- Line 278: “the size of the "high" sample was smaller than that of the "low" sample (37.9 278 nm vs. 13.1 nm, respectively” Did you mean mean/average size?
Yes and we have corrected it.
14- Section: 3.8. Visible spectrophotometry. More discussion is needed for flattening the absorption. It is well known that protein absorption is quite flat after 400 nm.
Yes, we extended the discussion.
15- Line 330: “whether the formation of the protein corona had an effect on the nature of the AuNPs.” Nature of AuNPs? Maybe you think properties and please specify which one.
Yes, we corrected the sentence. We wanted to check size and zeta potential of the nanoparticles.
16- Table 1 should contain units.
Yes, we added units.
17- Line 478: “made them a good candidate for in vivo analysis.” Made should be replaced with make.
Yes, we corrected that.

Reviewer 2 Report
The study gives a detailed understanding on the formation of protein corona on the surface of the AuNPs once it enters into the biological system. The manuscript is precise and all the relevant findings have been explained in detail. There are few suggestions/corrections below that have to be included in the manuscript before publication:
1. Change the title from “Evaluation the protein corona around the biologically produced gold nanoparticles” to “Evaluation of the protein corona around the biologically produced gold nanoparticles”.
2. Synthesis of gold nanoparticles using fungal species other than Fusarium oxysporum has been studied earlier (Verticillium sp- doi: 10.1002/1521-3773(20011001)40:19<3585::aid-anie3585>3.0.co;2-k.; Trichothecium sp- DOI: https://doi.org/10.1166/jbn.2005.012; Aspergillus niger- https://doi.org/10.1002/ceat.200800647). Can the author explain why they have chosen Fusarium and what are the advantages compared to other methods? Is it possible to obtain pure nanoparticles without any contamination of organic components? The authors should discuss this in the manuscript.
3. Include reference number within a square bracket or parentheses throughout the manuscript.
4. In section 1, Introduction, from line number 51-54, the authors cited papers about the biological production of silver nanoparticles. This seems irrelevant here, since the present study focuses on the gold nanoparticles. There are already several reports available on the biological production of AuNPs, hence the authors should include the above papers (comment 1) as references.
5. In section 3.5, Zetasizer analysis, line number 278-279, “the size of the "high" sample was smaller than that of the "low" sample (37.9 278 nm vs. 13.1 nm, respectively)”. The authors have misplaced the size value. Please check and revise the same.
6. In line number 282, italicize the number 3.6 in sub-section FTIR characterization.
7. In Figure 6B absorption spectra, keep the order same (graph insert) as in 6A (starting from lower concentration to higher concentration- high, high 10%, high 25%, high 55%).
8. The authors should include conflict of interest if any at the end of the manuscript.
9. The authors must include animal ethical committee approval in the manuscript.
Author Response
Dear Reviewer,
Thank you for your stimulating comments and suggestions. We have added your inputs to the text and I believe it significantly improved quality of the manuscript.
Our corrections are listed below in detail.
Best regards,
Veronika Benson
Comments
- Change the title from “Evaluation the protein corona around the biologically produced gold nanoparticles” to “Evaluation of the protein corona around the biologically produced gold nanoparticles”.
Yes, we corrected that.
- Synthesis of gold nanoparticles using fungal species other than Fusarium oxysporum has been studied earlier (Verticillium sp- doi: 10.1002/1521-3773(20011001)40:19<3585::aid-anie3585>3.0.co;2-k.; Trichothecium sp- DOI: https://doi.org/10.1166/jbn.2005.012; Aspergillus niger- https://doi.org/10.1002/ceat.200800647). Can the author explain why they have chosen Fusarium and what are the advantages compared to other methods? Is it possible to obtain pure nanoparticles without any contamination of organic components? The authors should discuss this in the manuscript.
We added the citations in the introduction part and discussed the organic content of the nanoparticles after Table 3.
- Include reference number within a square bracket or parentheses throughout the manuscript.
Reference style is corrected.
- In section 1, Introduction, from line number 51-54, the authors cited papers about the biological production of silver nanoparticles. This seems irrelevant here, since the present study focuses on the gold nanoparticles. There are already several reports available on the biological production of AuNPs, hence the authors should include the above papers (comment 1) as references.
Yes, we deleted those parts.
- In section 3.5, Zetasizer analysis, line number 278-279, “the size of the "high" sample was smallerthan that of the "low" sample (37.9 278 nm vs. 13.1 nm, respectively)”. The authors have misplaced the size value. Please check and revise the same.
Yes, thank you, we corrected that.
- 6. In line number 282, italicize the number 3.6 in sub-section FTIR characterization.
Yes, we corrected that.
- In Figure 6B absorption spectra, keep the order same (graph insert) as in 6A (starting from lower concentration to higher concentration- high, high 10%, high 25%, high 55%).
Yes, the graphs are corrected.
- The authors should include conflict of interest if any at the end of the manuscript.
Yes, we included a text regarding Conflicts of Interest.
- The authors must include animal ethical committee approval in the manuscript.
Yes, we added a text regarding the Animals’ procedures approval.
